# A Packet Collision Reduction Based on Reselection for LTE V2X Mode 4

Masashi Asano and Masahiro Fujii *,†

Division of Engineering and Agriculture Graduate School of Regional Development and Creativity, Utsunomiya University, Utsunomiya 321-8505, Japan
* Correspondence: fujii@is.utsunomiya-u.ac.jp
† Current address: 7-1-2 Yoto Utsunomiya, Japan.

**Abstract:** Vehicle to Everything (V2X) is a technology that includes communication between the vehicles and everything such as Vehicle to Vehicle (V2V), Vehicle to Infrastructure (V2I), and Vehicle to Pedestrian (V2P). Long Term Evolution (LTE) V2X based on LTE supports a sidelink communication in which User Equipment (UE) communicates with each other. In Mode 4 of the sidelink communication, the UE autonomously selects a radio resource that is not expected to be used by other UEs based on sensing information. However, a resource can be selected by simultaneous UEs and packet collisions occur because of the periodic resource reselection. In this paper, we propose two resource selection methods for the reselection using information originally included in the control information. Through computer simulations, we show that the proposed methods can improve the packet reception rate without requiring restrictions such as additional information. The main strength of this method is that it effectively utilizes the information contained in Resource Reservation Interval (RRI), which is used in the Semi-Persistent Scheduling wireless resource allocation algorithm. In this research, the value of RRI, which has been used in standards, is utilized to improve performance while maintaining compatibility. Since our method is designed under conditions that maintain compatibility with existing standards, it may or may not have a significant effect, but it does not degrade performance.

**Keywords:** V2X; V2V; LTE V2X; sidelink; Mode 4; SPS

## 1. Introduction

With the advance of Intelligent Transport Systems (ITS), Vehicle to Everything (V2X) communication has been attracting attention. V2X can be roughly divided into wireless LAN-based Dedicated Short Range Communication (DSRC) and cellular-based Cellular V2X (C-V2X). The former has been standardized as IEEE 802.11p [1] in the United States in 2010 by extending the IEEE 802.11 standard [2]. In Japan, ARIB STD-T109 [3] has been standardized by the Association of Radio Industries and Businesses (ARIB) in 2012 after formulating the experimental guidelines, ITS FORUM RC-006 [4]. On the other hand, C-V2X has been standardized as Long Term Evolution (LTE) V2X based on existing LTE by Third Generation Partnership (3GPP) release 14 in 2017. LTE V2X supports the uplink/downlink communication which corresponds to Vehicle to Network (V2N) and the sidelink communication, which corresponds to V2V, V2I, and V2P, as shown in Figure 1. Ref. [5] analyzes scenarios of the C-V2X and gives a survey of the related articles. Modes 3 and 4 are available as ways to use radio resources in the sidelink communication [6,7]. In Mode 3, an evolved Node B (eNB) allocates the resources to User Equipment (UE) under network coverage. In Mode 4, the UEs autonomously select the resources without the need for control by the eNB.

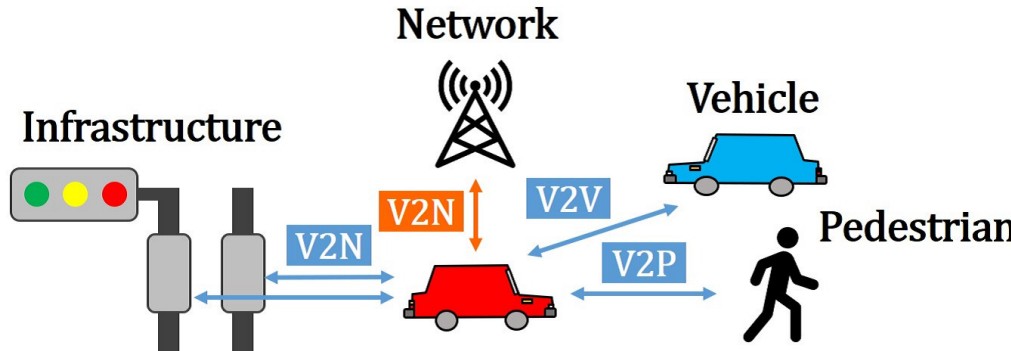

**Figure 1.** LTE V2X communication.

The UEs communicate with each other by sensing-based Semi-Persistent Scheduling (SPS) for LTE V2X Mode 4. For autonomous resource selection, packet collisions are reduced by selecting resources that are not expected to be used by other UEs from reservation information and a measurement of the received power. However, because the resource selection depends on past information, the resource newly selected by one UE cannot be detected by other UEs until transmission on the resource. Therefore, the packet collision may occur because the UEs that select a resource at the same time cannot estimate the resources selected by each other. Regarding an evaluation of the received power, the resource released at the reselection cannot be effectively utilized because the evaluated value is smoothed without information about the resource reselection of other UEs.

Basic performance evaluation of Mode 4 is verified in a highway scenario [8–11] and a urban scenario [12,13]. The evaluation of congestion control and the impact of parameters are presented in [14,15] and [16–18]. In addition to these works, analytical models are provided in [19,20]. On the other hand, methods to reduce the packet collisions are proposed in [21–31]. In [21–23], each of the UEs notifies other UEs of resources virtually reserved by the UE in advance. In [24], the UEs adjust the timing of reselection with other UEs. In [25], the hidden terminal problem is mitigated by sending additional acknowledgments and feeding this information back to the transmitter. As a similar approach to Mode 3, a cluster-based resource allocation has been proposed in [26]. However, these methods are not preferable from the viewpoint of compatibility with the standard because they require additional information and the format of control information needs to be changed. Geo-based resource allocation methods corresponding to the position of the UEs on a road have been proposed in [27–29], and can avoid packet collisions with the UEs located on Non-Line of Sight (NLOS). Even so, it is necessary to know a correspondence between the UE position and the resource pool in advance, and it is difficult to achieve a high degree of efficiency unless all UEs follow the same rule. In [30], although the UEs use alternately two different resources to reduce consecutive packet collisions, it is difficult to estimate the resources used by other UEs. A nonlinear averaging method has been proposed in [31], which assigns higher priority to the latest information in evaluating the received power to effectively utilize the resources released by other UEs. Nevertheless, the effect is not sufficient for the reselection that occurred immediately before evaluating the received power. An enhancement to the SPS algorithm has been introduced in which each vehicle announces the reservation information for the reselection in [32]. A balanced resource allocation scheme has been proposed in [33].

In this paper, we propose two resource reselection methods in consideration of other UE reselection, using only the information originally included in the control information [34]. In the proposed methods, the UEs can use the released resources by evaluating the received power only for the resources after other UEs' reselection. Furthermore, it can avoid packet collisions by selecting from the resource range not selected by other UEs. The main novelty of this paper is the effective use of information contained in the already operational Resource Reservation Interval (RRI) in the wireless resource allocation

algorithm using the SPS. Various methods have been studied to improve the performance of the wireless resource allocation problem by using additional information, but these methods require extensions that do not conform to the standard. In contrast, this study proposes and examines a new method for using the value of RRI included in the standard to improve performance while maintaining compatibility with the standard. The proposed method is highly feasible because it does not require any special extensions to existing standards. Since our method is designed to maintain compatibility with existing standards, it may or may not have a significant effect, but there are no disadvantages.

## 2. LTE V2X Mode 4

In the sidelink communication, the UEs transmit a packet in synchronization with each other by using the Global Navigation Satellite System (GNSS), eNB, and UEs [35]. The sidelink communication supports a repeated transmission up to two times using Hybrid Automatic Repeat Request (HARQ) as an option. For Mode 4, which does not require the eNB control, it is optimized to autonomously avoid the packet collisions for periodic transmissions.

### 2.1. Radio Resources

Figure 2 shows an example of the resource usage for sidelink communication. A radio resource consists of a subframe (1 ms) and subchannels divided between the time domain and frequency domain. A subchannel consists of multiple Resource Blocks (RBs) of 180 kHz. The figure shows an example when the number of the subchannel is 2, and each UE transmits using the resource represented by the rectangle. The UEs transmit control information and data using adjacent frequencies. The control information is transmitted as Sidelink Control Information (SCI) [36] over Physical Sidelink Control Channel (PSCCH). Although the SCI can notice periodic resource reservations, it cannot be used to reserve reselected resources. The data are transmitted in Transport Blocks (TBs) over Physical Sidelink Shared Channel (PSSCH). The TB is assumed to contain Cooperative Awareness Message (CAM) [37], Decentralized Environmental Notification Message (DENM) [38], and so on.

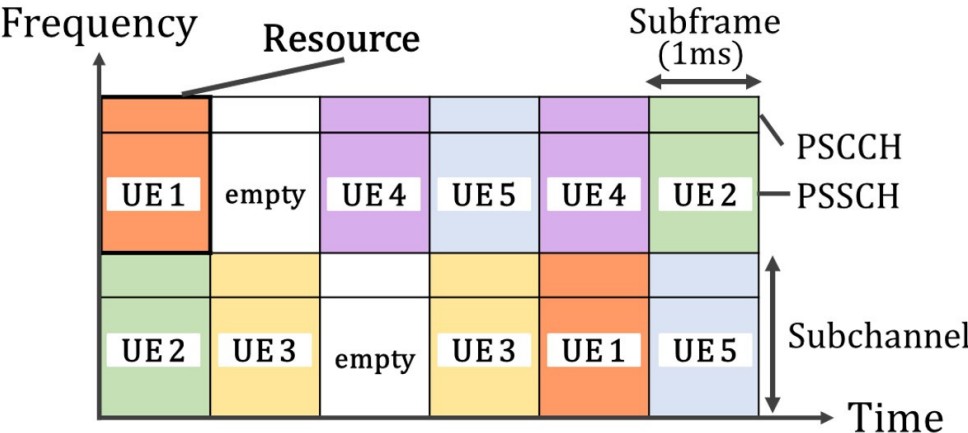

**Figure 2.** An example of resource usage for the sidelink communication.

The structure of SCI Format 1 used in Mode 3 and 4 is illustrated in Table 1. SCI Format 1 includes the following information:

- Priority: the priority of the packet set by upper layers;
- Resource Reservation: the time period of the intended next use of the resource;
- Frequency resource location of initial transmission and retransmission: Resource Indication Value (RIV) corresponding to a starting subchannel index and length in terms of contiguously allocated subchannels;

- Time gap between initial transmission and retransmission: subframe differences in repeated transmissions;
- Modulation and coding scheme: the combination of modulation scheme and code rate;
- Retransmission index: a boolean denoting whether the first transmission or second transmission.

The resource reservation value is interpreted for the other UEs as $X$ according to Table 2, where it is equal to RRI divided by 100. The RRI represents the transmission cycle, which is basically 100 ms, and 20, 50, 100, 200, . . . , 1000 ms can be configured [6].

**Table 1.** Structure of SCI Format 1.

| | |
|---|---|
| Priority | 3 bits |
| Resource Reservation | 4 bits |
| Frequency resource location of initial transmission and retransmission | Calculation based on the number of subchannels |
| Time gap between initial transmission and retransmission | 4 bits |
| Modulation and coding scheme | 5 bits |
| Retransmission index | 1 bit |
| Reserved information | add until 32 bit total |

**Table 2.** Determination of the Resource Reservation field in SCI Format 1.

| Resource Reservation Field in Value $X$ SCI Format 1 | Indicated | Condition |
|---|---|---|
| '0001', '0010', . . . , '1010' | Decimal equivalent of the field | The UE keeps the resources for the next transmission after $100 \times X$ ms |
| '1011' | 0.5 | The UE keeps the resources for the next transmission after 50 ms |
| '1100' | 0.2 | The UE keeps the resources for the next transmission after 20 ms |
| '0000' | 0 | The UE does not keep the resources for the next transmission |
| '1101', '1110', '1111' | Reserved | |

In the sidelink communication, the UEs transmit by Single Carrier Frequency Division Multiple Access (SC-FDMA) like LTE uplink. A subframe consists of 14 symbols that are time-multiplexed as shown in Figures 3 and 4, where the last symbol is used for Tx/Rx switching and timing adjustment [39]. Two DeModulation Reference Signals (DM-RS) are contained in a subframe in Release 12 as shown in Figure 3. To improve the demodulation accuracy in a fast-moving environment, the DM-RS is increased to four symbols in Release 14 as shown in Figure 4.

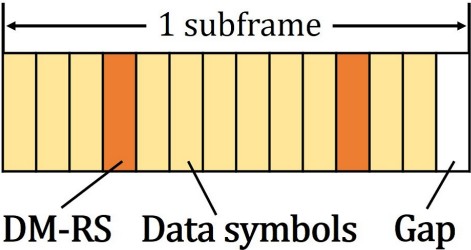

**Figure 3.** Subframe structure for sidelink for Release 12.

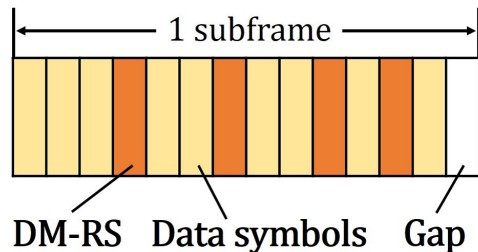

**Figure 4.** Subframe structure for sidelink for Release 14.

### 2.2. Sensing-Based SPS

In Mode 4, the UEs select a resource using the sensing-based SPS. The SPS is performed by sensing for the past one second, and resource reservation information is notified by PSCCH. The UEs set Reselection Counter (RC) depending on the RRI from a uniform distribution in the range of $[5, 15]$ for RRI $\geq 100$, $[10, 30]$ for RRI $= 50$, and $[25, 75]$ for RRI $= 20$. The RC is decremented every transmission. The UEs use the same subchannel resources until the RC reaches 0. In addition, when it is equal to 0, they decide whether to reselect the resource based on the probability $p_k$ of keeping their resource where $p_k$ can be set in the range $[0, 0.8]$. If the UE decides to reselect the resource, a selection window is set in the time interval $[n + T_1, n + T_2]$ for subframe $n$ where $T_1 \leq 4$ and $20 \leq T_2 \leq 100$. A candidate resource in selection window $R_{x,y}$ is defined with subchannel $x$ and subframe $y$ where $0 \leq x \leq$ thenumber of subchannels $- 1$ and $n + T_1 \leq y \leq n + T_2$. Then, the UE initializes the set $\mathbb{S}_A$ with all candidate resources and excludes them according to the following steps a, b, and c.

In Step a, the UE excludes unmonitored resources due to the constraint of half-duplex transmission. In Step b, the UE excludes any resources reserved by other UEs. When the packets are received with PSCCH Reference Signal Receive Power (PSSCH-RSRP) of the TB corresponding to the SCI greater than or equal to threshold $P_{th}$ and satisfy $m + P_{rsvp} = y$ where parameter $m$ is a subframe in sensing window defined by $[n - 1000, n - 1]$ and $P_{rsvp}$ is the RRI notified over PSCCH, they represent the reserved resources. When the resource is reselected, $P_{rsvp}$ is set to 0 because the UE does not make a reservation. $P_{th}$ is set by upper layers depending on priority $\alpha$ of the transmitter and $\beta$ of the receiver (both within 0 and 7). It can be in the range $[-128, -2]$ dB and can be expressed as [6,40]

$$P_{th} = -128 + 2(\alpha \times 8 + \beta) \tag{1}$$

After the exclusion by Steps a and b, the UE checks whether the remainder of $\mathbb{S}_A$ contains at least 20% of the total number of candidate resources. If this condition is not satisfied, then Step b is repeated with $P_{th}$ increased by 3 dB. In Step c, the UE excludes any resources whose received power is highly evaluated. Metric $E_{x,y}$ is evaluated as the linear average of Received Signal Strength Indicator (RSSI) that for RRI $\geq 100$ can be expressed as

$$E_{x,y} = \frac{1}{10} \sum_{i=1}^{10} \text{RSSI}\{R_{x,y-100 \times i}\} \tag{2}$$

The UE moves $R_{x,y}$ with the smallest $E_{x,y}$ from $\mathbb{S}_A$ to the set $\mathbb{S}_B$ initialized to an empty set. This step is repeated until the number of candidate resources in $\mathbb{S}_B$ reaches greater than or equal to 20 % of the total number of candidate resources. After these steps, the UE randomly selects one of the resources in $\mathbb{S}_B$.

An example of the exclusion process by Steps a, b, and c is demonstrated in Figure 5. In this figure, the UE excludes the candidate resources in subframe 1020 by Step a because the UE has transmitted in subframe 420. In addition, the UE excludes it in subframe 1050 and subchannel 0 because the UE receives a packet with $P_{rsvp} = 100$ in subframe 950 and subchannel 0. Similarly, the UE excludes it in subframe 1080 and subchannel 1 because the UE largely evaluates its RSSI.

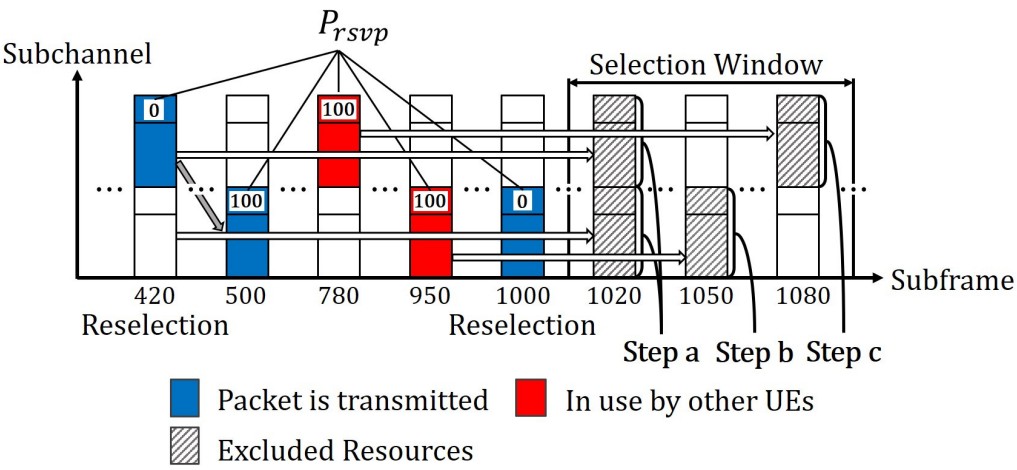

**Figure 5.** An example of exclusion by Steps a, b, and c.

### 2.3. Evaluating RSSI for Released Resources

In Step c, the UE evaluates the RSSI by the linear average in units of 100 subframes up to 1000 subframes in the past. If other UEs reselect a resource, the RSSI for the released resources may be greatly evaluated since the average value includes the resources that have been used before their reselection. In such a case, they are no longer candidates for the UE despite the resources released by other UEs. As a result, it results in inefficient use of resources.

Figure 6 shows an example of the RSSI evaluation for the released resources. This figure indicates that UE2 has reselected in subframe 820 and UE1 reselects in subframe 1000. Now, $R_{1,1020}$ is a candidate resource for UE1 corresponding to the resource released by UE2 in subframe 820. When evaluating the RSSI for $R_{1,1020}$, UE1 largely evaluates it because it includes the resources before UE2's reselection. $R_{1,1020}$ may not be able to be included in $\mathbb{S}_B$ since $\mathbb{S}_B$ is composed of resources evaluated to be small RSSI.

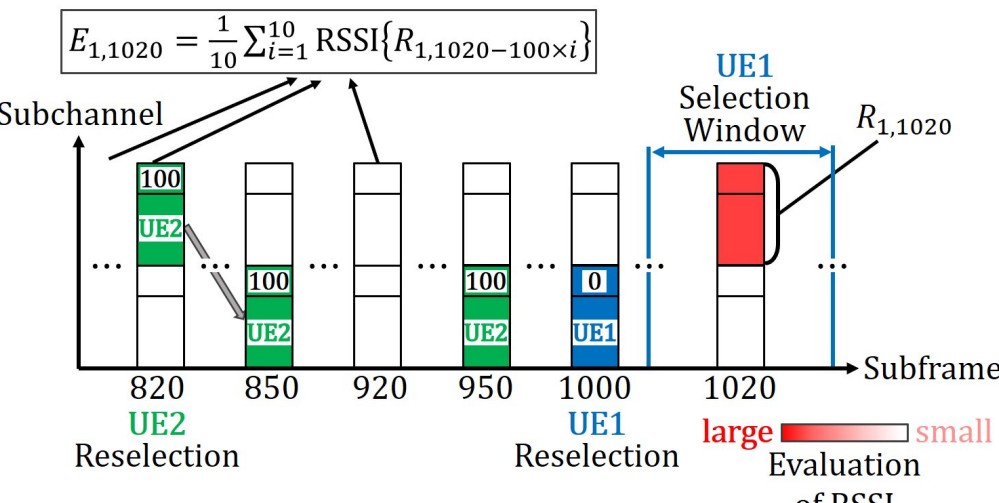

**Figure 6.** An example of evaluating RSSI for released resources.

*2.4. Overlapping Selection Window*

In a resource selection by the SPS, the UEs select a resource to avoid packet collisions with each other. The resources used by other UEs are estimated based on a feature of using the same subchannel resources periodically. Hence, if a UE reselects a resource, other UEs cannot detect which resource was selected until the transmission using the newly selected resource is performed.

Figure 7 shows an example of an overlapping selection window. This figure indicates that UE2 has reselected in subframe 950 and UE1 reselects in subframe 1000. The range of the selection window for UE2 is from a maximum of subframe 951 to 1050, and that for UE1 is from a maximum of subframe 1001 to 1100. In this case, the section between subframe 1001 and 1050 is overlapped, and UE1 may select the same resource as UE2.

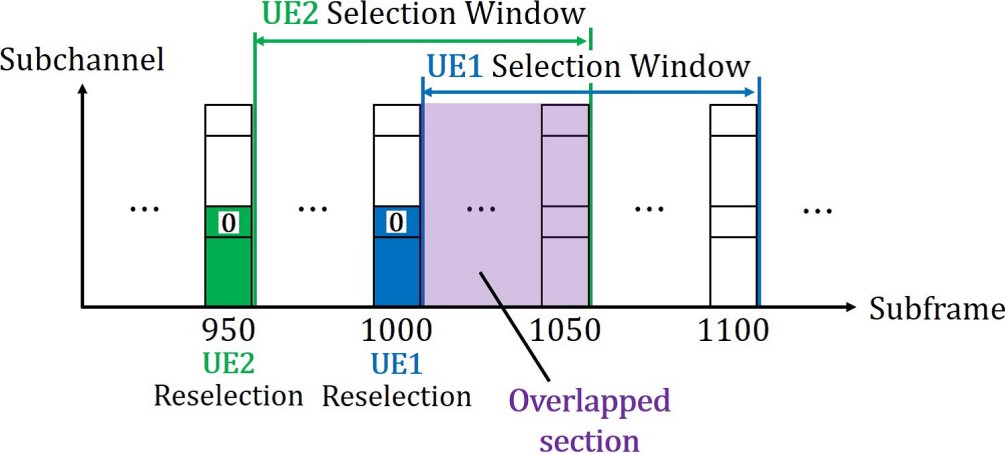

**Figure 7.** Example of overlapping selection window.

## 3. Proposed Method

In this paper, we propose new two resource selection methods in consideration of the fact that $P_{rsvp}$ notified over PSCCH reaches 0 when the UEs reselect. The proposed methods can reduce the packet collisions without constraints such as additional information.

### 3.1. Partial Averaging (PA) Method

We propose a Partial Averaging (PA) method to address the problem of large RSSI evaluations for the released resources. The PA method evaluates the RSSI only for resources after receiving a packet with $P_{rsvp} = 0$. This method enables the UEs to prevent the RSSI from being greatly evaluated by limiting the smoothing range to the resources and be able to make effective use of the released resources.

Figure 8 shows an example of the RSSI evaluation using the PA method. This figure indicates that UE2 has reselected in subframe 820 and UE1 reselects in subframe 1000. When evaluating the RSSI for $R_{1,1020}$ in the SPS, UE1 largely evaluates it because of averaging resources before subframe 820 that was used by UE2. In contrast, the PA method evaluates the RSSI only for resources after subframe 920 since the packet with $P_{rsvp} = 0$ has been received in subframe 820. In subframe 920, UE2 has not transmitted, and the RSSI for $R_{1,1020}$ is evaluated to be smaller. As a result, UE1 can make effective use of $R_{1,1020}$.

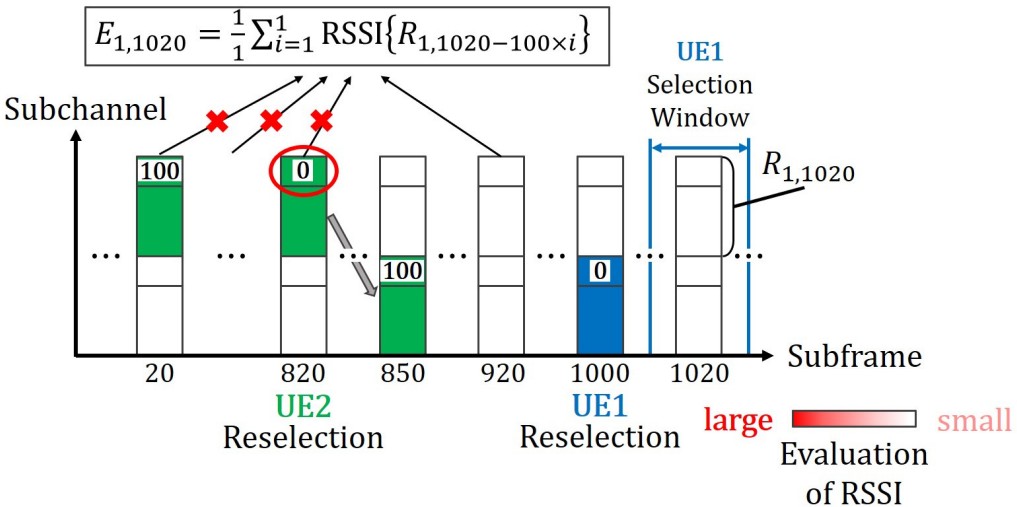

**Figure 8.** An example of the PA method.

### 3.2. Overlapping Avoidance (OA) Method

In this section, we propose an Overlapping Avoidance (OA) method to address the problem of the overlapping selection window. In the OA method, the UE first defines the subframe as $m'$ in which the last packet with $P_{rsvp} = 0$ was received within the last 100 subframes from the resource selection. Then, the UE moves $R_{x,y}^{\mathbb{S}_B}$, which is a subset of $\mathbb{S}_B$, to $\mathbb{S}_{B1}$ if $y < m' + 100$ and $\mathbb{S}_{B2}$ if $y \geq m' + 100$, respectively. Finally, the UE randomly selects a resource from $\mathbb{S}_{B2}$ if $|\mathbb{S}_{B2}| \neq 0$ and $\mathbb{S}_{B1}$ if $|\mathbb{S}_{B2}| = 0$. Now, $\mathbb{S}_{B1}$ and $\mathbb{S}_{B2}$ mean the set of resources that can be selected by the UE that transmits the packet with $P_{rsvp} = 0$ and the set of resources in which the UE has free from the possibility of the selection, respectively. Note that the selection window of the UE is up to 100 subframes, but the resources with $y = m' + 100$ are included in $\mathbb{S}_{B2}$ because the resources in $m' + 100$ are excluded by Step a.

Figure 9 shows an example using the OA method. This figure indicates that UE2 has reselected in subframe 950, and UE1 reselects in subframe 1000. In this case, $m' = 950$ because UE1 has received a packet with $P_{rsvp} = 0$ in subframe 950. UE1 divides $\mathbb{S}_B$ into $\mathbb{S}_{B1}$ and $\mathbb{S}_{B2}$, where $\mathbb{S}_{B1}$ includes the candidate resources between subframe 1001 and 1049, and $\mathbb{S}_{B2}$ includes the candidate resources between subframe 1050 and 1100. As a result, UE1 can avoid the same resource selection as UE2 by preferentially selecting a resource from $\mathbb{S}_{B2}$.

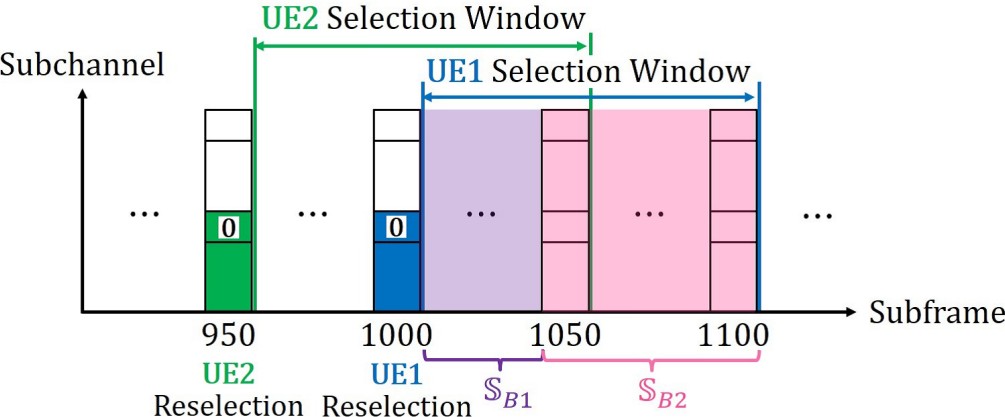

**Figure 9.** An example of the OA method.

## 4. Numerical Excamples

### 4.1. Simulation Environment

In this paper, we use the traffic flow simulator SUMO (Simulation of Urban MObility) [41] to simulate the behavior of the UEs and update the position of each UE every 100 ms. For the mobility simulation of UE traffic, a highway scenario is used as shown in Figure 10. The highway length is equal to 5 km, the lane width is equal to 4 m, and each way has three lanes. We use the central 2 km as the evaluation range.

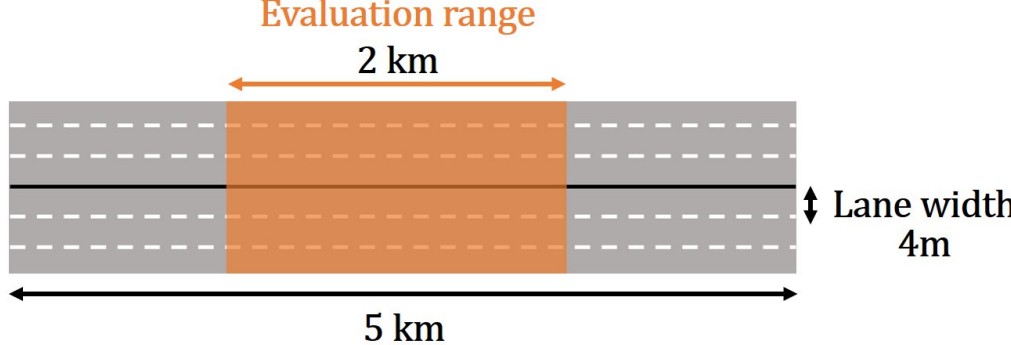

**Figure 10.** Highway scenario.

The received power is calculated using the WINNER+ B1 propagation model [42]. The propagation loss $PL$ [dB] in the WINNER+B1 model is calculated as follows:

- for $10 < d < d'_{BP}$:

$$PL = 22.7 \log(d) + 27 + 20 \log(f_c), \tag{3}$$

- for $d'_{BP} < d < 5000$:

$$PL = 40 \log(d) + 7.56 - 17.3 \log(h'_t) - 17.3 \log(h'_r) + 2.7 \log(f_c), \tag{4}$$

where $d$ is the distance between the transmitter and the receiver in meters, and $f_c$ is the carrier frequency in GHz. The effective antenna heights $h'_t$ and $h'_r$ are computed as follows: $h'_t = h_t - 1.0$ m, $h'_r = h_r - 1.0$ m, where $h_t$ and $h_r$ are the actual antenna heights at the transmitter and the receiver, respectively. The parameter $d'_{BP}$ is calculated as follows:

$$d'_{BP} = \frac{4h'_t h'_r f_c}{c} \tag{5}$$

where $c$ is the speed of light, and the value is 299792458 m/s in a vacuum. Although the speed of light decelerates slightly in air, the deceleration is usually negligible. $d'_{BP}$ means the breakpoint distance, and it is well known that the radio propagation has different characteristics after the breakpoint distance. Therefore, in Equations (3) and (4), the propagation loss must be divided into cases with $d'_{BP}$ as a boundary. In consideration of the effective range of the WINNER+ B1 model, if $d$ is less than 10 m, the propagation loss is calculated as $d = 10$ m. The path loss is used to calculate the Signal to Interference Noise Ratio (SINR) for each received packet. The packet error rate [43] based on the estimated SINR is used to determine the success or failure of packet reception. In this paper, channel fading and shadowing are not considered for the basic verification of the proposed methods.

The simulation parameters are set as listed in Table 3. These parameters are applicable for normal ITS radio propagation. Probability $p_k$ to keep the same resource is set to 0.0. This means that the reselection is always performed when the RC reaches 0. Selection window widths $T_1$ and $T_2$ are set so that the selection window is the maximum. Threshold $P_{th}$ is set to the lowest possible value. The bandwidth, carrier frequency, packet size, transmission power, noise figure, antenna height, and antenna gain are according to the 3GPP evaluation scenario [44]. The number of subchannels is set to 2 assuming Quadrature Phase Shift Keying (QPSK) and coding rate 1/2 for the bandwidth. The RRI is determined at the time of UE generation based on probability $p_g$ as shown in Tables 4–6, and it is not changed after the generation.

**Table 3.** Simulation parameters.

| Parameter | Value |
|---|---|
| $p_k$ | 0.0 |
| $T_1, T_2$ | 1, 100 |
| $P_{th}$ | −128 dBm |
| Bandwidth | 10 MHz |
| Carrier frequency | 6 GHz |
| Packet size | 300 bytes |
| Number of subchannels | 2 |
| Transmission power | 23 dBm |
| Noise figure | 9 dBm |
| Antenna height | 1.5 m |
| Antenna gain | 3 dBi |
| Number of UEs | 600 |

**Table 4.** Generation pattern 1.

| | RRI |
|---|---|
| | 100 |
| $p_g$ | 1.0 |

**Table 5.** Generation pattern 2.

| | RRI | | | | | |
|---|---|---|---|---|---|---|
| | 100 | 200 | 400 | 600 | 800 | 1000 |
| $p_g$ | 0.5 | 0.1 | 0.1 | 0.1 | 0.1 | 0.1 |

**Table 6.** Generation pattern 3.

| | RRI | | | | |
|---|---|---|---|---|---|
| | 100 | 200 | 300 | 400 | 500 |
| $p_g$ | 0.2 | 0.2 | 0.2 | 0.2 | 0.2 |

For performance evaluation, Packet Reception Rate (PRR) [44] is used. The PRR is calculated as follows:

$$\text{PRR}_{d_1,d_2} = \frac{\sum_{n=1}^{N} X_{d_1,d_2}^{n}}{\sum_{n=1}^{N} Y_{d_1,d_2}^{n}} \tag{6}$$

where $Y_{d_1,d_2}^{n}$ is the number of UEs located in the range $[d_1,d_2)$ from the transmitter for the $n$-th $(1 \geq n \geq N)$ packet, and $X_{d_1,d_2}^{n}$ is the number of UEs with successful reception among $Y_{d_1,d_2}^{n}$.

*4.2. Numerical Results*

Figure 11 shows a comparison between the conventional scheme and the proposed methods for generation pattern 1. "No interference" indicates the case of the interference-free, and this curve shows the upper bound of PRR.

The proposed PA method can obtain a higher PRR than the conventional scheme in a long range of each UE. We expect that the PA method can reduce packet collisions when the transmitter is far from the receiver. The PA method enables the UEs to increase the released resources to $\mathbb{S}_B$. Since $\mathbb{S}_B$ is composed of the resources in ascending order of the RSSI, it is possible to exclude the resources, which is expected to be used by other UEs and whose RSSI is evaluated to be small, from $\mathbb{S}_B$. In particular, other UEs that use the resource, evaluated to be small RSSI, are located far from the UE because the evaluation of the RSSI depends on only the distance between the UEs. On the other hand, the proposed OA method can obtain a higher PRR than the conventional scheme in the short range of each UE. We expect that the OA method can reduce packet collisions when the transmitter is at a short distance from the receiver. The OA method can avoid the same resource selection for the UE that transmits the packet with $P_{rsvp} = 0$. This is effective for UEs located at a short distance from the transmitter because it is possible to obtain $P_{rsvp}$ from the received packet easily. Fortunately, we can make use of the proposed PA and OA methods at the same time without additional information and specific constraints. The combination method obtains higher PRR than the conventional scheme in the short and long range of each UE. Because the two methods interact with each other, it is possible to obtain a high PRR in any range. In addition, each proposed method always obtains the same or better performance as the conventional scheme. This is because the proposed method operates in the same way as the conventional scheme if there is no notification of $P_{rsvp} = 0$.

Figures 12 and 13 show a comparison between the conventional and the proposed methods for generation pattern 2 and 3. These results have the same tendency of each method in comparison with Figure 11, and the combination of the two methods can obtain the highest performance. However, the performance of the conventional scheme approaches "No interference", and the improvement in performance by the proposed method is decreasing accordingly. This is caused by decreased frequency of resource usage decreases due to a mixture of UEs with several RRIs. In addition, although the proposed method is effective when the resource is reselected, low frequency results in small improvement.

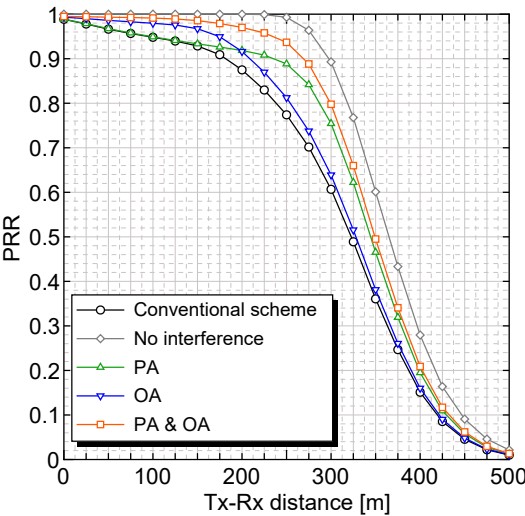

**Figure 11.** Effect of the PRR for generation pattern 1.

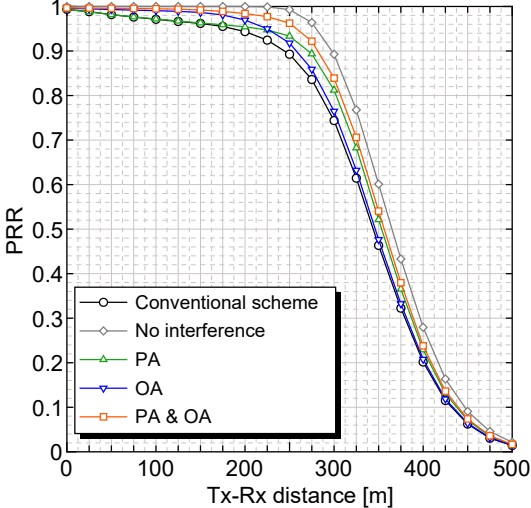

**Figure 12.** Effect of the PRR for generation pattern 2.

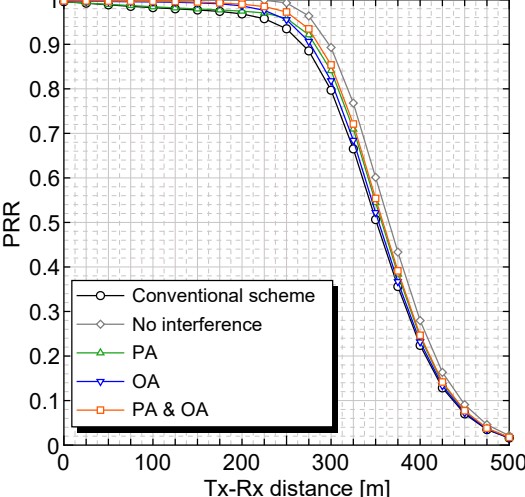

**Figure 13.** Effect of the PRR for generation pattern 3.

## 5. Conclusions

In this paper, we proposed two methods that contributed to the performance improvement without additional information for LTE V2X Mode 4. The proposed methods can reduce the packet collisions by using the property that $P_{rsvp}$ reaches 0 when the resource is reselected. We evaluated the effectiveness of the proposed methods with three RRI patterns. The proposed PA method made effective use of the released resources and was highly effective in the long range. The proposed OA method avoided the overlapping selection window and was highly effective in the short range. The combination of the PA and OA methods outperformed the conventional scheme in all ranges due to the interaction of the two methods. Furthermore, no deterioration in performance was confirmed. The main contribution of this paper is to make effective use of the information contained in $P_{rsvp}$ used in the SPS wireless resource allocation algorithm. Although various methods have been studied to improve the performance of the wireless resource allocation problem by using additional information, these methods require extensions that do not conform to the standard. In contrast, this research utilizes $P_{rsvp}$ that have been used in standards to improve performance while maintaining compatibility. The greatest advantage of this method is that it can be introduced without modifying existing standards. We also propose two methods for utilizing $P_{rsvp}$, the PA and OA, and their fusion method, which always outperforms the performance of each of them alone. Our method is designed to maintain compatibility with existing standards, so it may or may not have a significant effect, but there are no disadvantages.

For future work, we are studying the effectiveness of the proposed methods in different scenarios such as effects of the channel fading and shadowing, UE density, and road model including the NLOS environment.

**Author Contributions:** Writing—original draft, M.A.; Writing—review & editing, M.F. All authors have read and agreed to the published version of the manuscript.

**Funding:** This research received no external funding.

**Institutional Review Board Statement:** Not applicable.

**Informed Consent Statement:** Not applicable.

**Data Availability Statement:** Not applicable.

**Conflicts of Interest:** The authors declare no conflict of interest.

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
