# Peer review of "A Packet Collision Reduction Based on Reselection for LTE V2X Mode 4"

_applsci, doi:10.3390/app12178733_

Round 1
Reviewer 1 Report
The introduction to this paper is concise and easy to read. However, the introduction of the method is too brief, and its parameters cannot be explained in detail. For example, what is the difference between ht and hr? C is the speed of light? (what kind of speed of which kind of light?) Why do we calculate d'BP parameter for? It is suggested that the authors can establish a scene of use for help describe and use the parameters listed or within the method, which can make this manuscript and method more feasible.
Reviewer 2 Report
This paper proposes two methods, using information that was originally included in the management information for re-selection. Computer simulations show that the proposed methods can improve the packet receiving rate without requiring additional limitations. The article addresses an important issue, but there are a number of areas that could be reviewed and significantly improved. Some specific comments:
1. The abstract must be improved; it must be clearer. Consider reviewing the abstract and highlighting the novelty, major findings, and conclusions.
2. The introduction should be extended with the most important findings. The actuality and novelty of the current paper must be clarified. What research gap did you find from previous researchers in your field? Mention it at the end of the Introduction section. It will improve the strength of the article.
3. The literature review is not current and must be expanded. Only a few references for 2020 were found (one of the references is provided by the authors). The list of references should be supplemented with a few recent articles that briefly mention current trends and developments related to the manuscript.
4. Have you validated your modeling against the physical realization? This is probably beyond the scope of this article, but physical validation would make the results much stronger.
5. The conclusion section of the paper is very condensed. If the authors' conclusions provided a more interesting contribution to the motivation for this study and some interesting open questions for future research, it would encourage the reader to take more interest in the topic.
In the opinion of the reviewer, the manuscript has the potential to be published but would need improvements and revisions.
Round 2
Reviewer 2 Report
The authors have greatly improved the article and answered most of the questions. Congratulations to the authors.